# Representational Continuity for Unsupervised Continual Learning

**Divyam Madaan**[1*]  **Jaehong Yoon**[2,3 †]  **Yuanchun Li**[5,6]  **Yunxin Liu**[5,6]  **Sung Ju Hwang**[2,4]
New York University[1]    KAIST[2]    Microsoft Research[3]    AITRICS[4]
Institute for AI Industry Research (AIR)[5]    Tsinghua University[6]
divyam.madaan@nyu.edu, {jaehong.yoon,sjhwang82}@kaist.ac.kr
liyuanchun@air.tsinghua.edu.cn, liuyunxin@air.tsinghua.edu.cn

## Abstract

Continual learning (CL) aims to learn a sequence of tasks without forgetting the previously acquired knowledge. However, recent CL advances are restricted to supervised continual learning (SCL) scenarios. Consequently, they are not scalable to real-world applications where the data distribution is often biased and unannotated. In this work, we focus on *unsupervised continual learning (UCL)*, where we learn the feature representations on an unlabelled sequence of tasks and show that reliance on annotated data is not necessary for continual learning. We conduct a systematic study analyzing the learned feature representations and show that unsupervised visual representations are surprisingly more robust to catastrophic forgetting, consistently achieve better performance, and generalize better to out-of-distribution tasks than SCL. Furthermore, we find that UCL achieves a smoother loss landscape through qualitative analysis of the learned representations and learns meaningful feature representations. Additionally, we propose **L**ifelong **U**nsupervised **M**ixu**p** (Lump), a simple yet effective technique that interpolates between the current task and previous tasks' instances to alleviate catastrophic forgetting for unsupervised representations. We release our code online.

## 1 Introduction

Recently continual learning (Thrun, 1995) has gained a lot of attention in the deep learning community due to its ability to continually learn on a sequence of non-stationary tasks (Kumar & Daume III, 2012; Li & Hoiem, 2016) and close proximity to the human learning process (Flesch et al., 2018). However, the inability of the learner to prevent forgetting of the knowledge learnt from the previous tasks has been a long-standing problem (McCloskey & Cohen, 1989; Goodfellow et al., 2013). To address this problem, a large body of methods (Rusu et al., 2016; Zenke et al., 2017; Yoon et al., 2018; Li et al., 2019; Aljundi et al., 2019; Buzzega et al., 2020) have been proposed; however, all these methods focus on the supervised learning paradigm, but obtaining high-quality labels is expensive and often impractical in real-world scenarios. In contrast, CL for unsupervised representation learning has received limited attention in the community. Although Rao et al. (2019) instantiated a continual unsupervised representation learning framework (Curl), it is not scalable for high-resolution tasks, as it is composed of MLP encoders/decoders and a simple MoG generative replay. This is evident in their limited empirical evaluation using digit-based gray-scale datasets.

Meanwhile, a set of directions have shown huge potential to tackle the representation learning problem without labels (He et al., 2020; Chen et al., 2020a; Grill et al., 2020; Chen et al., 2020b; Chen & He, 2021; Zbontar et al., 2021) by aligning contrastive pairs of training instances or maximizing the similarity between two augmented views of each image. However, a common assumption for existing methods is the availability of a large amount of unbiased and unlabelled datasets to learn the feature representations. We argue that this assumption is not realistic for most of the real-time applications, including self-driving cars (Bojarski et al., 2016), medical applications (Kelly et al., 2019) and conversational agents (Li et al., 2020). The collected datasets are often limited in size during the initial training phase (Finn et al., 2017), and datasets/tasks change continuously with time.

---

[*]Corresponding author. [†] The work was done while the author was an intern at Microsoft Research.

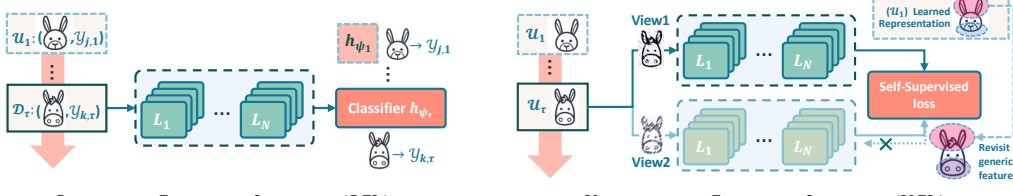

Figure 1: **Illustration of supervised and unsupervised continual learning.** The objective of SCL is to learn the ability to classify labeled images in the current task while preserving the past tasks' knowledge, where the tasks are non-iid to each other. On the other hand, UCL aims to learn the representation of images without the presence of labels and the model learns general-purpose representations during sequential training.

To accommodate such continuous shifts in data distributions, representation learning models need to increment the knowledge without losing the representations learned in the past. With this motivation, we attempt to bridge the gap between unsupervised representation learning and continual learning to address the challenge of learning the representations on a sequence of tasks. Specifically, we focus on *unsupervised continual learning (UCL)*, where the goal of the continual learner is to learn the representations from a stream of unlabelled data instances without forgetting (see Figure 1). To this end, we extend various existing SCL strategies to the unsupervised continual learning framework and analyze the performance of current state-of-the-art representation learning techniques: *SimSiam* (Chen & He, 2021) and *BarlowTwins* (Zbontar et al., 2021) for UCL. Surprisingly, we observe that the unsupervised representations are comparatively more robust to catastrophic forgetting across all datasets and simply finetuning on the sequence of tasks can outperform various state-of-the-art continual learning alternatives. Furthermore, we show that UCL generalize better to various out of distribution tasks and outperforms SCL for few-shot training scenarios (Section 5.2).

We demystify the robustness of unsupervised representations by investigating the feature similarity, measured by centered kernel alignment (CKA) (Kornblith et al., 2019) between two independent UCL and SCL models and between UCL and SCL models. We notice that two unsupervised model representations have a relatively high feature similarity compared to two supervised representations. Furthermore, in all cases, two models have high similarity in lower layers indicating that they learn similar low-level features. Further, we measure the $\ell_2$ distance between model parameters (Neyshabur et al., 2020) and visually compare the feature representations learned by different SCL and UCL strategies. We observe that UCL obtains human perceptual feature patterns for previous tasks, demonstrating their effectiveness to alleviate catastrophic forgetting (Section 5.3). We conjecture that this is due to their characteristic ability to learn general-purpose features (Doersch et al., 2020), which makes them transfer better and comparatively more robust to catastrophic forgetting.

To gain further insights, we visualize the loss landscape (Li et al., 2018) of the UCL and SCL models and observe that UCL obtains a flatter and smoother loss landscape than SCL. Additionally, we propose a simple yet effective technique coined **L**ifelong **U**nsupervised **M**ixup (LUMP), which utilizes mixup (Zhang et al., 2018) for unlabelled training instances. In particular, LUMP interpolates between the current task examples and examples from previous instances to minimize catastrophic forgetting. We emphasize that LUMP is easy to implement, does not require additional hyperparameters, and simply trains on the interpolated instances. To this end, LUMP requires little, or no modification to existing rehearsal-based methods effectively minimizes catastrophic forgetting even with uniformly selecting the examples from replay buffer. We show that LUMP with UCL outperforms the state-of-the-art supervised continual learning methods across multiple experimental settings with significantly lower catastrophic forgetting. In summary, our contributions are as follows:

- We attempt to bridge the gap between continual learning and representation learning and tackle the two crucial problems of continual learning with unlabelled data and representation learning on a sequence of tasks.

- Systematic quantitative analysis shows that UCL achieves better performance over SCL with significantly lower catastrophic forgetting on Sequential CIFAR-10, CIFAR-100, and Tiny-ImageNet. Additionally, we evaluate out-of-distribution tasks and few-shot training demonstrating the expressive power of unsupervised representations.

- We provide visualization of the representations and loss landscapes, which show that UCL learns discriminative, human perceptual patterns and achieves a flatter and smoother loss landscape. Furthermore, we propose **L**ifelong **U**nsupervised **M**ixup (LUMP) for UCL, which effectively alleviates catastrophic forgetting and provides better qualitative interpretations.

## 2 RELATED WORK

**Continual learning.** We can partition the existing continual learning methods into three categories. The *regularization* approaches (Li & Hoiem, 2016; Zenke et al., 2017; Schwarz et al., 2018; Ahn et al., 2019) impose a regularization constraint to the objective that mitigates catastrophic forgetting. The *architectural* approaches (Rusu et al., 2016; Yoon et al., 2018; Li et al., 2019) avoid this problem by including task-specific parameters and allowing the expansion of the network during continual learning. The *rehearsal* approaches (Rebuffi et al., 2017; Rolnick et al., 2019; Aljundi et al., 2019) allocate a small memory buffer to store and replay the examples from the previous task. However, all these methods are confined to supervised learning, which limits their application in real-life problems. Rao et al. (2019); Smith et al. (2021) tackled the problem of continual unsupervised representation learning; however, their methods are restricted to simple low-resolution tasks and not scalable to large-scale continual learning datasets.

**Representational learning.** A large number of works have addressed the unsupervised learning problem in the standard machine learning framework. Specifically, contrastive learning frameworks (He et al., 2020; Chen et al., 2020a; Grill et al., 2020; Chen et al., 2020b;c) that learn the representations by measuring the similarities of positive and negative pairs have gained a lot of attention in the community. However, all these methods require large batches and negative sample pairs, which restrict the scalability of these networks. Chen & He (2021) tackled these limitations and proposed *SimSiam*, that use standard Siamese networks (Bromley et al., 1994) with the stop-gradient operation to prevent the collapsing of Siamese networks to a constant. Recently, Zbontar et al. (2021) formulated an objective that pushes the cross-correlation matrix between the embeddings of distorted versions of a sample closer to the identity matrix. However, all these methods assume the presence of large datasets for pre-training, which is impractical in real-world applications. In contrast, we tackle the problem of incremental representational learning and learn the representations sequentially while maximizing task adaptation and minimizing catastrophic forgetting.

## 3 PRELIMINARIES

### 3.1 PROBLEM SETUP

We consider the continual learning setting, where we learn on a continuum of data consisting of $T$ tasks $\mathcal{T}_{1:T} = (\mathcal{T}_1 \ldots \mathcal{T}_T)$. In supervised continual learning, each task consists a task descriptor $\tau \in \{1 \ldots T\}$ and its corresponding dataset $\mathcal{D}_\tau = \{(\boldsymbol{x}_{i,\tau}, y_{i,\tau})_{i=1}^{n_\tau}\}$ with $n_\tau$ examples. Each input-pair $(\boldsymbol{x}_{i,\tau}, y_{i,\tau}) \in \mathcal{X}_\tau \times \mathcal{Y}_\tau$, where $(\mathcal{X}_\tau, \mathcal{Y}_\tau)$ is an unkown data distribution. Let us consider a network $f_\Theta : \mathcal{X}_\tau \to \mathbb{R}^D$ parametrized by $\Theta = \{\boldsymbol{w}_l\}_{l=1}^{l=L}$, where $\mathbb{R}^D$ and $L$ denote $D$-dimensional embedding space and number of layers respectively. The classifier is denoted by $h_\psi : \mathbb{R}^D \to \mathcal{Y}_\tau$. The network error using cross entropy loss (CE) for SCL with finetuning can be formally defined as:

$$\mathcal{L}_{\text{SCL}}^{\text{FINETUNE}} = \text{CE}\left(h_\psi\left(f_\Theta\left(\boldsymbol{x}_{i,\tau}\right), \tau\right), y_{i,\tau}\right). \tag{1}$$

In this work, we assume the absence of label supervision during training and focus on unsupervised continual learning. In particular, each task consists of $\mathcal{U}_\tau = \{(\boldsymbol{x}_{i,\tau})_{i=1}^{n_\tau}\}$, $\boldsymbol{x}_{i,\tau} \in \mathcal{X}_\tau$ with $n_\tau$ examples. Our aim is to learn the representations $f_\Theta : \mathcal{X}_\tau \to \mathbb{R}^D$ on a sequence of tasks while preserving the knowledge of the previous tasks. We introduce the representation learning framework and propose LUMP in Section 4 to learn unsupervised representations while effectively mitigating catastrophic forgetting.

### 3.2 LEARNING PROTOCOL AND EVALUATION METRICS

Currently, the traditional continual learning strategies follow the standard training protocol, where we learn the network representations $f_\Theta : \mathcal{X}_\tau \to \mathcal{Y}_\tau$ on a sequence of tasks. In contrast, our goal is to learn the feature representations $f_\Theta : \mathcal{X}_\tau \to \mathbb{R}^D$, so we follow a two-step learning protocol to obtain the model predictions. First, we pre-train the representations on a sequence of tasks $T_{1:T} = (\mathcal{T} \ldots \mathcal{T}_T)$ to obtain the representations. Next, we evaluate the quality of our pre-trained representations using a K-nearest neighbor (KNN) classifier (Wu et al., 2018) following the setup in Chen et al. (2020a); Chen & He (2021); Zbontar et al. (2021).

To validate knowledge transfer of the learned representations, we adopt the metrics from the SCL literature (Chaudhry et al., 2019b; Mirzadeh et al., 2020). Let $a_{\tau,i}$ denote the test accuracy of task $i$ after learning task $\mathcal{T}_\tau$ using a KNN on frozen pre-trained representations on task $\mathcal{T}_\tau$. More formally, we can define the metrics to evaluate the continually learned representations as follow:

1. **Average accuracy** is the average test accuracy of all the tasks completed until the continual learning of task $\tau$: $A_\tau = \frac{1}{\tau} \sum_{i=1}^{\tau} a_{\tau,i}$

2. **Average Forgetting** is the average performance decrease of each task between its maximum accuracy and accuracy at the completion of training: $F = \frac{1}{T-1} \sum_{i=1}^{T-1} \max_{\tau \in \{1,\ldots,T\}} (a_{\tau,i} - a_{T,i})$

## 4 UNSUPERVISED CONTINUAL LEARNING

### 4.1 CONTINUOUS REPRESENTATION LEARNING WITH SEQUENTIAL TASKS

To learn feature representations, contrastive learning (Chen et al., 2020a;b; He et al., 2020) maximizes the similarity of representations between the images of the same views (positive pairs) and minimizes the similarity between images of different views (negative pairs). However, these methods require large batches, negative sample pairs (Chen et al., 2020a;b), or architectural modifications (He et al., 2020; Chen et al., 2020c), or non-differentiable operators (Caron et al., 2020), which makes their application difficult for continual learning scenarios. In this work, we focus on SimSiam (Chen & He, 2021) and BarlowTwins (Zbontar et al., 2021), which tackle these limitations and achieve state-of-the-art performance on standard representation learning benchmarks.

**SimSiam (Chen & He, 2021)** uses a variant of Siamese networks (Bromley et al., 1994) for learning input data representations. It consists of an encoder network $f_\Theta$, which is composed of a backbone network, and is shared across a projection MLP and prediction MLP head $h(\cdot)$. Specifically, SimSiam minimizes the cosine-similarity between the output vectors of the projector and the predictor MLP across two different augmentations for an image. Initially, we consider FINETUNE, which is a a naive CL baseline that minimizes the cosine-similarity between the projector output ($z_{i,\tau}^1 = f_\Theta(x_{i,\tau}^1)$) and the predictor output ($p_{i,\tau}^2 = h(f_\Theta(x_{i,\tau}^2))$) on a sequence of tasks as follows:

$$\mathcal{L}_{\text{UCL}}^{\text{FINETUNE}} = \frac{1}{2} D(p_{i,\tau}^1, \text{stopgrad}(z_{i,\tau}^2)) + \frac{1}{2} D(p_{i,\tau}^2, \text{stopgrad}(z_{i,\tau}^1)), \qquad (2)$$

$$\text{where } D(p_{i,\tau}^1, z_{i,\tau}^2) = -\frac{p_{i,\tau}^1}{\|p_{i,\tau}^2\|_2} \cdot \frac{z_{i,\tau}^2}{\|z_{i,\tau}^2\|_2},$$

$x_{i,\tau}^1$ and $x_{i,\tau}^2$ are two randomly augmented views of an input example $x_{i,\tau} \in \mathcal{T}_\tau$ and $\|\cdot\|_2$ denotes the $\ell_2$-norm. Note that, the $\text{stopgrad}$ is a crucial component in SimSiam to prevent the trivial solutions obtained by Siamese networks. Due to its simplicity and effectiveness, we chose Simsiam to analyze the performance of unsupervised representations for continual learning.

**BarlowTwins (Zbontar et al., 2021)** minimizes the redundancy between the embedding vector components of the distorted versions of an instance while conserving the maximum information inspired from Barlow (1961). In particular, the objective function eliminates the SimSiam $\text{stopgrad}$ component and instead makes the cross-correlation matrix computed between the outputs of two identical networks closer to the identity matrix. Let $\mathcal{C}$ be the cross-correlation matrix between the outputs of two Siamese branches along the batch dimension and $Z_1$ and $Z_2$ denote the batch embeddings of the distorted views for all images of a batch from the current task ($x_\tau \in \mathcal{U}_\tau$). The objective function for UCL with finetuning and BarlowTwins can then be defined as:

$$\mathcal{L}_{\text{UCL}}^{\text{FINETUNE}} = \sum_i (1 - \mathcal{C}_{ii})^2 + \lambda \cdot \sum_i \sum_{j \neq i} \mathcal{C}_{ij}^2, \quad \text{where } \mathcal{C}_{ij} = \frac{\sum_{\mathcal{B}} z_{\mathcal{B},i}^1 z_{\mathcal{B},j}^2}{\sqrt{\sum_{\mathcal{B}} \left(z_{\mathcal{B},i}^1\right)^2} \sqrt{\sum_{\mathcal{B}} \left(z_{\mathcal{B},j}^2\right)^2}}. \qquad (3)$$

$\lambda$ is a positive constant trading off the importance of the invariance and redundancy reduction terms of the loss, $i$ and $j$ denote the network's output vector dimensions. Similar to SimSiam, BarlowTwins is simple, easy to implement, and can be applied to existing continual learning strategies with little or no modification.

## 4.2 PRESERVING REPRESENTATIONAL CONTINUITY: A VIEW OF EXISTING SCL METHODS

Learning feature representations from labelled instances on a sequence of tasks has been long studied in continual learning. However, the majority of these learning strategies are not directly applicable to UCL. To compare with the regularization-based strategies, we extend *Synaptic Intelligence (SI)* (Zenke et al., 2017) to UCL and consider the online per-synapse consolidation during the entire training trajectory of the unsupervised representations. For architectural-based strategies, we investigate *Progressive Neural Networks (PNN)* (Rusu et al., 2016) and learn the feature representations progressively using the representations learning frameworks proposed in Section 4.1.

We also formulate *Dark Experience Replay (DER)* (Buzzega et al., 2020) for UCL. DER for SCL alleviates catastrophic forgetting by matching the network logits across a sequence of tasks during the optimization trajectory. Notably, the loss for SCL-DER can be defined as follow:

$$\mathcal{L}_{\text{SCL}}^{\text{DER}} = \mathcal{L}_{\text{SCL}}^{\text{FINETUNE}} + \alpha \cdot \mathbb{E}_{(x,p) \sim \mathcal{M}} \big[ \|\text{softmax}(p) - \text{softmax}(h_\psi(x_{i,\tau}))\|_2^2 \big], \tag{4}$$

where $p = h_{\psi_\tau(x)}$, $\mathcal{L}_{\text{SCL}}^{\text{FINETUNE}}$ denotes the cross-entropy loss on the current task (see Equation (1)) and random examples are selected using reservoir sampling from the replay-buffer $\mathcal{M}$. Since, we do not have access to the labels for UCL, we cannot minimize the aforementioned objective.

Instead, we utilize the output of the projected output by the backbone network to preserve the knowledge of the past tasks over the entire training trajectory. In particular, DER for UCL consists of a combination of two terms. The first term learns the representations using SimSiam from Equation (2) or BarlowTwins from Equation (3) and the second term minimizes the Euclidean distance between the projected outputs to minimize catastrophic forgetting. More formally, UCL-DER minimizes the following loss:

$$\mathcal{L}_{\text{UCL}}^{\text{DER}} = \mathcal{L}_{\text{UCL}}^{\text{FINETUNE}} + \alpha \cdot \mathbb{E}_{(x) \sim \mathcal{M}} \big[ \|f_{\Theta_\tau}(x) - f_\Theta(x_{i,\tau})\|_2^2 \big] \tag{5}$$

However, the performance of the rehearsal-based methods is sensitive to the choice of $\alpha$ and often requires supervised training setup, task identities, and boundaries. To tackle this issue, we propose Lifelong Unsupervised Mixup in the subsequent subsection, which interpolates between the current and past task instances to mitigate catastrophic forgetting effectively.

## 4.3 LIFELONG UNSUPERVISED MIXUP

The standard Mixup (Zhang et al., 2018) training constructs virtual training examples based on the principle of Vicinal Risk Minimization . In particular, let $(x_i, y_i)$ and $(x_j, y_j)$ denote two random feature-target pairs sampled from the training data distribution and let $(\tilde{x}, \tilde{y})$ denote the interpolated feature-target pair in the vicinity of these examples; mixup then minimizes the following objective:

$$\mathcal{L}^{\text{MIXUP}}(\tilde{x}, \tilde{y}) = \text{CE}\left(h_\psi\left(f_\Theta\left(\tilde{x}\right)\right), \tilde{y}\right), \tag{6}$$
$$\text{where } \tilde{x} = \lambda \cdot x_i + (1 - \lambda) \cdot x_j \text{ and } \tilde{y} = \lambda \cdot y_i + (1 - \lambda) \cdot y_j.$$

$\lambda \sim \text{Beta}(\alpha, \alpha)$, for $\alpha \in (0, \infty)$. In this work, we focus on lifelong self-supervised learning and propose Lifelong Unsupervised Mixup (LUMP) that utilizes mixup for UCL by incorporating the instances stored in the replay-buffer from the previous tasks into the vicinal distribution. In particular, LUMP interpolates between the examples of the current task $(x_{i,\tau}) \in \mathcal{U}_\tau$ and random examples selected using uniform sampling from the replay buffer, which encourages the model to behave linearly across a sequence of tasks. More formally, LUMP minimizes the objective in Equation (2) and Equation (3) on the following interpolated instances $\tilde{x}_{i,\tau}$ for the current task $\tau$:

$$\tilde{x}_{i,\tau} = \lambda \cdot x_{i,\tau} + (1 - \lambda) \cdot x_{j,\mathcal{M}}, \tag{7}$$

where $x_{j,\mathcal{M}} \sim \mathcal{M}$ denotes the example selected using uniform sampling from replay buffer $\mathcal{M}$. The interpolated examples not only augments the past tasks' instances in the replay buffer but also approximates a regularized loss minimization (Zhang et al., 2021). During UCL, LUMP enhances the robustness of learned representation by revisiting the attributes of the past task that are similar to the current task. Recently, Kim et al. (2020); Lee et al. (2021); Verma et al. (2021); Shen et al. (2022) also employed mixup for contrastive learning. Our work is different from these existing works in that our objective is different, and we focus on unsupervised continual learning. To this end, LUMP successively mitigates catastrophic forgetting and learns discriminative & human-perceptual features over the current state-of-the-art SCL strategies (see Table 1 and Figure 4).

## 5 EXPERIMENTS

### 5.1 EXPERIMENTAL SETUP

**Baselines.** We compare with multiple supervised and unsupervised continual learning baselines across different categories of continual learning methods.

1. **Supervised continual learning.** FINETUNE is a vanilla supervised learning method trained on a sequence of tasks without regularization or episodic memory and MULTITASK optimizes the model on complete data. For regularization-based CL methods, we compare against SI (Zenke et al., 2017) and AGEM (Chaudhry et al., 2019a). We include PNN (Rusu et al., 2016) for architecture-based methods. Lastly, we consider GSS (Aljundi et al., 2019) that populates the replay-buffer using solid-angle minimization and DER (Buzzega et al., 2020) matches the network logits sampled through the optimization trajectory for rehearsal during continual learning.

2. **Unsupervised continual learning.** We consider the unsupervised variants of various SCL baselines to show the utility of the unsupervised representations for sequential learning. Specifically, we use SIMSIAM (Chen & He, 2021) and BARLOWTWINS (Zbontar et al., 2021), which are the state-of-the-art representational learning techniques for learning the unsupervised continual representations. We compare with FINETUNE and MULTITASK following the supervised learning baselines, and SI (Zenke et al., 2017), PNN (Rusu et al., 2016) for unsupervised regularization and architecture CL methods respectively. For rehearsal-based method, we compare with the UCL variant of DER (Buzzega et al., 2020) described in Section 4.2

**Datasets.** We compare the performance of SCL and UCL on various continual learning benchmarks using single-head ResNet-18 (He et al., 2016) architecture. **Split CIFAR-10** (Krizhevsky, 2012) consists of two random classes out of the ten classes for each task. **Split CIFAR-100** (Krizhevsky, 2012) consists of five random classes out of the 100 classes for each task. **Split Tiny-ImageNet** is a variant of the ImageNet dataset (Deng et al., 2009) containing five random classes out of the 100 classes for each task with the images sized $64 \times 64$ pixels.

**Training and evaluation setup.** We follow the hyperparameter setup of Buzzega et al. (2020) for all the SCL strategies and tune them for the UCL representation learning strategies. All the learned representations are evaluated with KNN classifier (Wu et al., 2018) across three independent runs. Further, we use the hyper-parameters obtained by SimSiam for training UCL strategies with BarlowTwins to analyze the sensitivity of UCL to hyper-parameters and for a fair comparison between different methods. We train all the UCL methods for 200 epochs and evaluate with the KNN classifier (Wu et al., 2018). We provide the hyper-parameters in detail in Table A.5.

### 5.2 QUANTITATIVE RESULTS

**Evaluation on SimSiam.** Table 1 shows the evaluation results for supervised and unsupervised representations learnt by SimSiam (Chen & He, 2021) across various continual learning strategies. In all cases, continual learning with unsupervised representations achieves significantly better performance than supervised representations with substantially lower forgetting. For instance, SI with UCL obtains better performance and $68\%$, $54\%$, and $44\%$ lower forgetting relative to the best-performing SCL strategy on Split CIFAR-10, Split CIFAR-100, and Split Tiny-ImageNet, respectively. Surprisingly, FINETUNE with UCL achieves higher performance and significantly lower forgetting in comparison to all SCL strategies except DER. Furthermore, LUMP improves upon the UCL strategies: $2.8\%$ and $5.9\%$ relative increase in accuracy and $15\%$ and $57.1\%$ relative decrease in forgetting on Split CIFAR-100 and Split Tiny-ImageNet, respectively.

**Evaluation on BarlowTwins.** To verify that unsupervised representations are indeed more robust to catastrophic forgetting, we train BarlowTwins (Zbontar et al., 2021) on a sequence of tasks. We notice that the representations learned with BarlowTwins substantially improve the accuracy and forgetting over SCL: $71.4\%$, $69.7\%$ and $73.2\%$ decrease in forgetting with FINETUNE on Split CIFAR-10, Split CIFAR-100 and Split Tiny-ImageNet respectively. Similarly, we observe that SI, and DER are more robust to catastrophic forgetting; however, PNN underperforms on complicated tasks since feature accumulation using adaptor modules is insufficient to construct useful representations for current task adaptation. Interestingly, representations learnt with BarlowTwins achieve lower forgetting for FINETUNE, DER and LUMP than SimSiam with comparable accuracy across all the datasets.

Table 1: **Accuracy and forgetting** of the learnt representations on Split CIFAR-10, Split CIFAR-100 and Split Tiny-ImageNet on Resnet-18 architecture with KNN classifier (Wu et al., 2018). All the values are measured by computing mean and standard deviation across three trials. The best and second-best results are highlighted in **bold** and underline respectively.

| METHOD | SPLIT CIFAR-10 | | SPLIT CIFAR-100 | | SPLIT TINY-IMAGENET | |
|---|---|---|---|---|---|---|
| | ACCURACY | FORGETTING | ACCURACY | FORGETTING | ACCURACY | FORGETTING |
| SUPERVISED CONTINUAL LEARNING | | | | | | |
| FINETUNE | 82.87 (± 0.47) | 14.26 (± 0.52) | 61.08 (± 0.04) | 31.23 (± 0.41) | 53.10 (± 1.37) | 33.15 (± 1.22) |
| PNN (Rusu et al., 2016) | 82.74 (± 2.12) | — | 66.05 (± 0.86) | — | 64.38 (± 0.92) | — |
| SI (Zenke et al., 2017) | 85.18 (± 0.65) | 11.39 (± 0.77) | 63.58 (± 0.37) | 27.98 (± 0.34) | 44.96 (± 2.41) | 26.29 (± 1.40) |
| A-GEM (Chaudhry et al., 2019a) | 82.41 (± 1.24) | 13.82 (± 1.27) | 59.81 (± 1.07) | 30.08 (± 0.91) | 60.45 (± 0.24) | 24.94 (± 1.24) |
| GSS (Aljundi et al., 2019) | 89.49 (± 1.75) | 7.50 (± 1.52) | 70.78 (± 1.67) | 21.28 (± 1.52) | 70.96 (± 0.72) | 14.76 (± 1.22) |
| DER (Buzzega et al., 2020) | 91.35 (± 0.46) | 5.65 (± 0.35) | 79.52 (± 1.88) | 12.80 (± 1.47) | 68.03 (± 0.85) | 17.74 (± 0.65) |
| MULTITASK | 97.77 (± 0.15) | — | 93.89 (± 0.78) | — | 91.79 (± 0.46) | — |
| UNSUPERVISED CONTINUAL LEARNING | | | | | | |
| SIMSIAM FINETUNE | 90.11 (± 0.12) | 5.42 (± 0.08) | 75.42 (± 0.78) | 10.19 (± 0.37) | 71.07 (± 0.20) | 9.48 (± 0.56) |
| PNN (Rusu et al., 2016) | 90.93 (± 0.22) | — | 66.58 (± 1.00) | — | 62.15 (± 1.35) | — |
| SI (Zenke et al., 2017) | **92.75** (± 0.06) | 1.81 (± 0.21) | 80.08 (± 1.30) | 5.54 (± 1.30) | 72.34 (± 0.42) | 8.26 (± 0.64) |
| DER (Buzzega et al., 2020) | 91.22 (± 0.30) | 4.63 (± 0.26) | 77.27 (± 0.30) | 9.31 (± 0.09) | 71.90 (± 1.44) | 8.36 (± 2.06) |
| LUMP | 91.00 (± 0.40) | 2.92 (± 0.53) | **82.30** (± 1.35) | 4.71 (± 1.52) | **76.66** (± 2.39) | 3.54 (± 1.04) |
| MULTITASK | 95.76 (± 0.08) | — | 86.31 (± 0.38) | — | 82.89 (± 0.49) | — |
| BARLOWTWINS FINETUNE | 87.72 (± 0.32) | 4.08 (± 0.56) | 71.97 (± 0.54) | 9.45 (± 1.01) | 66.28 (± 1.23) | 8.89 (± 0.66) |
| PNN (Rusu et al., 2016) | 87.52 (± 0.33) | — | 57.93 (± 2.98) | — | 48.70 (± 2.59) | — |
| SI (Zenke et al., 2017) | 90.21 (± 0.08) | 2.03 (± 0.22) | 75.04 (± 0.63) | 7.43 (± 0.67) | 56.96 (± 1.48) | 17.04 (± 0.89) |
| DER (Buzzega et al., 2020) | 88.67 (± 0.24) | 2.41 (± 0.26) | 73.48 (± 0.53) | 7.98 (± 0.29) | 68.56 (± 1.47) | 7.87 (± 0.44) |
| LUMP | 90.31 (± 0.30) | **1.13** (± 0.18) | 80.24 (± 1.04) | **3.53** (± 0.83) | 72.17 (± 0.89) | **2.43** (± 1.00) |
| MULTITASK | 95.48 (± 0.14) | — | 87.16 (± 0.52) | — | 82.42 (± 0.74) | — |

Figure 2: **Evaluation on Few-shot training** for Split CIFAR-100 across different number of training instances per task. The results are measured across three independent trials.

Figure 3: **CKA Feature similarity** between two independent UCL models (red), two independent SCL models (blue), and UCL and SCL model (green) for different strategies on Split CIFAR-100 test distribution.

**Evaluation on Few-shot training.** Figure 2 compares the effect of few-shot training on UCL and SCL, where each task has a limited number of training instances. Specifically, we conduct the experimental evaluation using 100, 200, 500, and 2500 training instances for each task in split CIFAR-100 dataset. Surprisingly, we observe that the gap in average accuracy between SCL and UCL methods widens with a decrease in the number of training instances. Note that UCL decreases the accuracy by $15.78\%p$ on average with lower forgetting when the number of training instances decreases from 2500 to 100; whereas, SCL obtains a severe $32.21\%p$ deterioration in accuracy. We conjecture that this is an outcome of the discriminative feature embeddings learned by UCL, which discriminates all the images in the dataset and captures more than class-specific information as also observed in Doersch et al. (2020). Furthermore, LUMP improves the performance over all the baselines with a significant margin across all few-shot experiments.

**Evaluation on OOD datasets.** We evaluate the learnt representations on various out-of-distribution (OOD) datasets in Table 2 to measure their generalization to unseen data distributions. In particular, we conduct the OOD evaluation on MNIST (LeCun, 1998), Fashion-MNIST (FMNIST) (Xiao et al., 2017), SVHN (Netzer et al., 2011), CIFAR-10 and CIFAR-100 (Krizhevsky, 2012) using a KNN classifier (Wu et al., 2018). We observe that unsupervised representations outperform the supervised representations in all cases across all the datasets. In particular, the UCL representations learned with Simsiam, and SI on Split-CIFAR-10 improves the absolute performance over the best-performing SCL strategy by 4.58%, 6.09%, 15.26%, and 17.07% on MNIST, FMNIST, SVHN, and CIFAR-100 respectively. Further, LUMP trained on Split-CIFAR-100 outperforms SI across all datasets and obtains comparable performance with Split CIFAR-10 dataset.

Table 2: **Comparison of accuracy** on out of distribution datasets using a KNN classifier (Wu et al., 2018) on pretrained SCL and UCL representations. We consider MNIST (LeCun, 1998), Fashion-MNIST (FMNIST) (Xiao et al., 2017), SVHN (Netzer et al., 2011) as out of distribution for Split CIFAR-100 and Split CIFAR-10. All the values are measured by computing mean and standard deviation across three trials. The best and second-best results are highlighted in **bold** and underline respectively.

| IN-CLASS | SPLIT CIFAR-10 | | | | SPLIT CIFAR-100 | | | |
|---|---|---|---|---|---|---|---|---|
| OUT-OF-CLASS | MNIST | FMNIST | SVHN | CIFAR-100 | MNIST | FMNIST | SVHN | CIFAR-10 |
| SUPERVISED CONTINUAL LEARNING | | | | | | | | |
| FINETUNE | 86.42 (± 1.11) | 74.47 (± 0.84) | 41.00 (± 0.85) | 17.42 (± 0.96) | 75.02 (± 3.97) | 62.37 (± 3.20) | 38.05 (± 0.73) | 39.18 (± 0.83) |
| SI (Zenke et al., 2017) | 87.08 (± 0.79) | 76.41 (± 0.81) | 42.62 (± 1.31) | 19.14 (± 0.91) | 79.96 (± 2.63) | 63.71 (± 1.36) | 40.92 (± 1.64) | 40.41 (± 1.71) |
| A-GEM (Chaudhry et al., 2019a) | 86.07 (± 1.94) | 74.74 (± 3.21) | 37.77 (± 3.49) | 16.11 (± 0.38) | 77.56 (± 3.21) | 64.16 (± 2.29) | 37.48 (± 1.73) | 37.91 (± 1.33) |
| GSS (Aljundi et al., 2019) | 70.36 (± 3.54) | 69.20 (± 2.51) | 33.11 (± 2.26) | 18.21 (± 0.39) | 76.54 (± 0.46) | 65.31 (± 1.72) | 35.72 (± 2.37) | 49.41 (± 1.81) |
| DER (Buzzega et al., 2020) | 80.32 (± 1.91) | 70.49 (± 1.54) | 41.48 (± 2.76) | 17.72 (± 0.25) | 87.71 (± 2.23) | 75.97 (± 1.29) | 50.26 (± 0.95) | 59.07 (± 1.06) |
| MULTITASK | 88.79 (± 1.13) | 79.50 (± 0.52) | 41.26 (± 1.95) | 27.68 (± 0.66) | 92.29 (± 3.37) | 86.12 (± 1.87) | 54.94 (± 1.77) | 54.04 (± 3.68) |
| UNSUPERVISED CONTINUAL LEARNING | | | | | | | | |
| *SimSiam* FINETUNE | 89.23 (± 0.99) | 80.05 (± 0.34) | 49.66 (± 0.81) | 34.52 (± 0.12) | 85.99 (± 0.86) | 76.90 (± 0.11) | 50.09 (± 1.41) | 57.15 (± 0.96) |
| SI (Zenke et al., 2017) | 93.72 (± 0.58) | 82.50 (± 0.51) | 57.88 (± 0.16) | 36.21 (± 0.69) | 91.50 (± 1.26) | 80.57 (± 0.93) | 54.07 (± 2.73) | 60.55 (± 2.54) |
| DER (Buzzega et al., 2020) | 88.35 (± 0.82) | 79.33 (± 0.62) | 48.83 (± 0.55) | 30.68 (± 0.36) | 87.96 (± 2.04) | 76.21 (± 0.63) | 47.70 (± 0.94) | 56.26 (± 0.16) |
| LUMP | 91.03 (± 0.22) | 80.78 (± 0.88) | 45.18 (± 1.57) | 31.17 (± 1.83) | 91.76 (± 1.17) | 81.61 (± 0.45) | 50.13 (± 0.71) | 63.00 (± 0.53) |
| MULTITASK | 90.69 (± 0.13) | 80.65 (± 0.42) | 47.67 (± 0.45) | 39.55 (± 0.18) | 90.35 (± 0.24) | 81.11 (± 1.86) | 52.20 (± 0.61) | 70.19 (± 0.15) |
| *BarlowTwins* FINETUNE | 86.86 (± 1.62) | 78.37 (± 0.74) | 44.64 (± 2.39) | 28.03 (± 0.52) | 76.08 (± 2.86) | 76.82 (± 0.83) | 42.95 (± 0.90) | 53.12 (± 0.13) |
| SI (Zenke et al., 2017) | 90.31 (± 0.69) | 80.58 (± 0.68) | 49.18 (± 0.51) | 31.80 (± 0.4) | 85.24 (± 0.99) | 78.82 (± 0.67) | 45.18 (± 1.37) | 53.99 (± 0.56) |
| DER (Buzzega et al., 2020) | 85.15 (± 2.19) | 77.96 (± 0.59) | 45.68 (± 0.93) | 27.83 (± 0.86) | 78.08 (± 1.95) | 76.67 (± 0.68) | 44.58 (± 1.01) | 53.24 (± 0.82) |
| LUMP | 88.73 (± 0.54) | 81.69 (± 0.45) | 51.53 (± 0.41) | 31.53 (± 0.36) | 90.22 (± 1.39) | 81.28 (± 0.91) | 50.24 (± 0.95) | 60.76 (± 0.87) |
| MULTITASK | 88.63 (± 1.38) | 79.49 (± 0.29) | 49.24 (± 2.44) | 36.33 (± 0.29) | 86.98 (± 1.70) | 79.40 (± 1.10) | 50.19 (± 0.81) | 49.50 (± 0.38) |

## 5.3 QUALITATIVE ANALYSIS

**Similarity in feature and parameter space.** We analyze the similarity between the representations learnt between (i) Two independent UCL models, (ii) Two independent SCL models (iii) SCL and UCL models using centered kernel alignment (CKA) (Kornblith et al., 2019) in Figure 3, which provides a score between 0 and 1 measuring the similarity between a pair of hidden representations. For two representations $\Theta_1 : \mathcal{X} \to \mathbb{R}^{d_1}$ and $\Theta_2 : \mathcal{X} \to \mathbb{R}^{d_1}$, $\text{CKA}(\Theta_1, \Theta_2) = \frac{||\text{Cov}(\Theta_1(x), \Theta_2(x))||_F^2}{||\text{Cov}(\Theta_1(x))||_F \cdot ||\text{Cov}(\Theta_2(x))||_F}$, where covariances are with respect to the test distribution. Additionally, we measure the $\ell_2$ distance (Neyshabur et al., 2020) between the parameters of two independent UCL models (see Table 3) and two independent SCL models (see Table 4). First, we observe that the representations learned by two independent UCL methods have a high feature similarity and lower $\ell_2$ distance compared to the two independent SCL methods, demonstrating UCL representations' robustness. Second, we note that the representations between any two independent models are highly similar in the lower layers indicating that they learn similar high-level features, including edges and shapes; however, the features are dissimilar for the higher modules. Lastly, we see that the representations between a UCL and SCL model are similar in the lower layers but diverge in the higher layers across all CL strategies.

**Visualization of feature space.** Next, we visualize the learned features to dissect further the representations learned by UCL and SCL strategies. Figure 4 shows the visualization of the latent feature maps for tasks $\mathcal{T}_0$ and $\mathcal{T}_{13}$ after the completion of continual learning. For $\mathcal{T}_0$, we observe that the SCL methods are prone to catastrophic forgetting, as the features appear noisy and do not have coherent patterns. In contrast, the features learned by UCL strategies are perceptually relevant and robust to catastrophic forgetting, with LUMP learning the most distinctive features. Similar to $\mathcal{T}_0$, we observe that the UCL features are more relevant and distinguishable than SCL for $\mathcal{T}_{13}$. Note that we randomly selected the examples and feature maps for all visualizations.

**Loss landscape visualization.** To gain further insights, we visualize the loss landscape of task $\mathcal{T}_0$ after the completion of training on task $\mathcal{T}_0$ and $\mathcal{T}_{19}$ for various UCL and SCL strategies in Figure 5. We measure the cross-entropy loss for all methods with a randomly initialized linear classifier for a fair evaluation of two different directions. We use the visualization tool from Li et al. (2018) that searches the task loss surface by repeatedly adding random perturbations to model weights. We observe that the loss landscape after $\mathcal{T}_0$ looks quite similar across all the strategies since the forgetting does not exist yet. However, after training $\mathcal{T}_{19}$, there is a clear difference with the UCL strategies obtaining a flatter and smoother loss landscape because UCL methods are more stable and robust to the forgetting, which hurts the loss landscapes of past tasks for SCL. It is important to observe that LUMP obtains a smoother landscape than other UCL strategies, demonstrating its effectiveness. We defer further analyses for feature and loss landscape visualization to Appendix A.2.

Table 3: $\ell_2$ distance between UCL parameters after completion of training.

| MODEL | FINETUNE | SI | DER | MULTITASK |
|---|---|---|---|---|
| FINETUNE | 60.00 (± 1.70) | | | |
| SI | 76.46 (± 0.48) | 92.35 (± 0.61) | | |
| DER | 55.60 (± 1.42) | 75.54 (± 0.97) | 48.76 (± 1.54) | |
| MULTITASK | 61.32 (± 0.59) | 79.95 (± 0.40) | 57.90 (± 0.86) | 61.42 (± 0.78) |

Table 4: $\ell_2$ distance between SCL parameters after completion of training.

| MODEL | FINETUNE | SI | DER | MULTITASK |
|---|---|---|---|---|
| FINETUNE | 183.31 (± 0.10) | | | |
| SI | 206.16 (± 0.28) | 226.05 (± 0.13) | | |
| DER | 202.61 (± 0.46) | 224.78 (± 0.75) | 219.06 (± 0.27) | |
| MULTITASK | 258.12 (± 0.26) | 277.30 (± 0.69) | 271.48 (± 0.45) | 314.84 (± 0.92) |

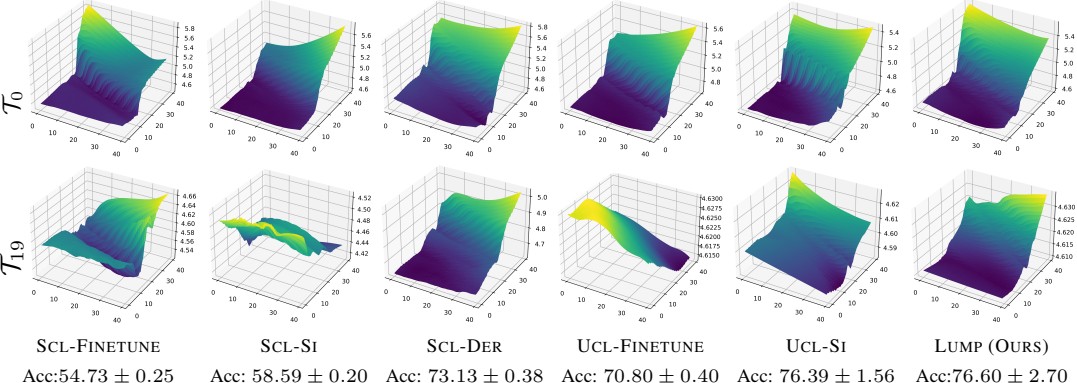

Figure 4: **Visualization of feature maps** for the second block representations learnt by SCL and UCL strategies (with Simsiam) for ResNet-18 architecture after the completion of CL for Split CIFAR-100 dataset ($n = 20$).

Figure 5: **Loss landscape visualization** of $\mathcal{T}_0$ after the completion of training on task $\mathcal{T}_0$ (**top**) and $\mathcal{T}_{19}$ (**bottom**) for Split CIFAR-100 dataset on ResNet-18 architecture. We use Simsiam for UCL methods.

## 6 DISCUSSION AND CONCLUSION

This work attempts to bridge the gap between unsupervised representation learning and continual learning. In particular, we establish the following findings for unsupervised continual learning.

**Surpassing supervised continual learning.** Our empirical evaluation across various CL strategies and datasets shows that UCL representations are more robust to catastrophic forgetting than SCL representations. Furthermore, we notice that UCL generalizes better to OOD tasks and achieves stronger performance on few-shot learning tasks. We propose *Lifelong unsupervised mixup (*LUMP*)*, which interpolates the unsupervised instances between the current task and past task and obtains higher performance with lower catastrophic forgetting across a wide range of tasks.

**Dissecting the learned representations.** We conduct a systematic analysis to understand the differences between the representations learned by UCL and SCL strategies. By investigating the similarity between the representations, we observe that UCL and SCL strategies have high similarities in the lower layers but are dissimilar in the higher layers. We also show that UCL representations learn coherent and discriminative patterns and smoother loss landscape than SCL.

**Limitations and future work.** In this work, we do not consider the high-resolution tasks for CL. We intend to evaluate the forgetting of the learnt representations on ImageNet (Deng et al., 2009) in future work, since UCL shows lower catastrophic forgetting and representation learning has made significant progress on ImageNet over the past years. In follow-up work, we intend to conduct further analysis to understand the behavior of UCL and develop sophisticated methods to continually learn unsupervised representations under various setups, such as class-incremental or task-agnostic CL.

ACKNOWLEDGEMENTS

We thank the anonymous reviewers for their insightful comments and suggestions. This work was supported by Microsoft Research Asia, the Engineering Research Center Program through the National Research Foundation of Korea (NRF) funded by the Korean Government MSIT (NRF-2018R1A5A1059921), Institute of Information & communications Technology Planning & Evaluation (IITP) grant funded by the Korea government (MSIT) (No.2019-0-00075, Artificial Intelligence Graduate School Program (KAIST) and 2021-0-01696). Any opinions, findings, and conclusions or recommendations expressed in this material are those of the authors and do not necessarily reflect the views of the funding agencies.

AUTHOR CONTRIBUTIONS

Divyam Madaan conceived of the presented idea, developed the experimental framework, carried out OOD evaluation, CKA visualization and took the lead in writing the manuscript. Jaehong Yoon performed the hyperparameter search, carried out the visualization of loss landscape and feature maps and performed the few-shot training analysis. Yuanchun Li, Yunxin Liu, and Sung Ju Hwang supervised the project.

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

# A    SUPPLEMENTARY MATERIAL

**Organization.** In the supplementary material, we provide the implementation details followed by the hyper-parameter configurations in Appendix A.1. Further, we show the other experiments we conducted and additional visualizations and results in Appendix A.2.

## A.1    EXPERIMENTAL DETAILS

**Implementations.** We use the DER (Buzzega et al., 2020) open-source codebase[1] for all the experiments. In particular, we reproduce all their experimental results for supervised continual learning and use various models with their set of hyper-parameters as our baselines. We follow the original representations for SimSiam[2] and BarlowTwins[3] for unsupervised continual learning. We verify our implementation by reproducing the reported results on CIFAR-10 in the original paper, where we train the representations on the complete CIFAR-10 dataset and evaluate on the test-set using KNN classifier (Wu et al., 2018). In particular, (Wu et al., 2018) stores the features for each instance in the task-level training set in a discrete memory bank. The optimal feature-level embeddings are then learned by instance-level discrimination, which maximally scatters the features of the training samples. Following prior works in representation learning, we use the task-level training set without any augmentation in the task-incremental setup for the supervised and unsupervised KNN evaluation.

**Hyperparameter configurations.** We use the tuned hyper-parameters reported by Buzzega et al. (2020) for all the SCL experiments. On the other hand, we tune the hyper-parameters for continual learning strategies for UCL. We provide the hyper-parameters setup for UCL for different datasets in Table A.5. We train all the UCL methods with a batch size of 256 for 200 epochs, while training the SCL methods with a batch size of 32 for 50 epochs following Buzzega et al. (2020). We observed that training the SCL methods further lead to a degredation in performance for all the methods. We use the same set of augmentations for both SCL and UCL except that we use `RandomResizedCrop` with scale in $[0.2, 1.0]$ for UCL (Wu et al., 2018; Chen & He, 2021) and `RandomCrop` for SCL. For rehearsal-based methods, we use the buffer size 200 for Split CIFAR-10, Split CIFAR-100 and 256 for Split Tiny-ImageNet dataset. We use a learning rate of 0.03 for SGD optimizer with weight decay 5e-4 and momentum 0.9.

Table A.5: **Hyperparameter configurations** for all the datasets on ResNet-18 architecture.

| METHOD | SPLIT CIFAR-10 | SPLIT CIFAR-100 | SEQ. TINY-IMAGENET |
|---|---|---|---|
| SI | $c : 100 \quad \xi : 1$ | $c : 0.1 \quad \xi : 1$ | $c : 0.01 \quad \xi : 1$ |
| PNN | $wd : 64$ | $wd : 12$ | $wd : 8$ |
| DER | $\alpha : 0.1$ | $\alpha : 0.1$ | $\alpha : 0.01$ |
| LUMP | $\lambda : 0.1$ | $\lambda : 0.1$ | $\lambda : 0.4$ |

## A.2    ADDITIONAL EXPERIMENTS

We provide additional loss landscape on Split CIFAR-100 in Figure A.6 and Figure A.7, Figure A.8 show the second and third block feature visualizations on Split CIFAR-100 respectively. Figure A.9 shows the feature visualizations for Split Tiny-ImageNet on ResNet-18 architecture.

---

[1] https://github.com/aimagelab/mammoth
[2] https://github.com/facebookresearch/simsiam
[3] https://github.com/facebookresearch/barlowtwins

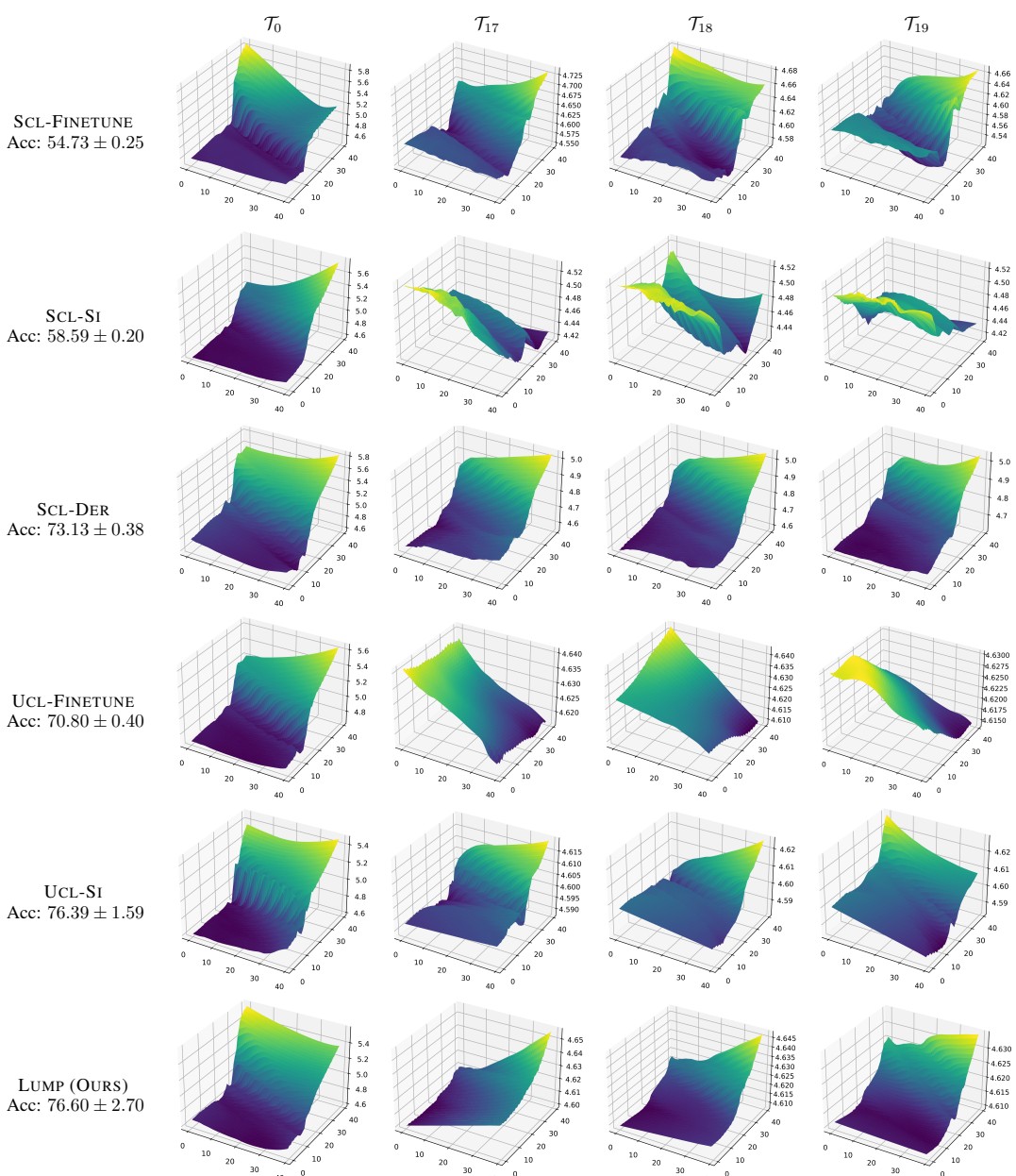

Figure A.6: **Loss landscape visualization** of $\mathcal{T}_0$ after the completion of training on task $\mathcal{T}_0$, $\mathcal{T}_{17}$, $\mathcal{T}_{18}$, and $\mathcal{T}_{19}$ for Split CIFAR-100 dataset on ResNet-18 architecture. We use Simsiam for UCL methods.

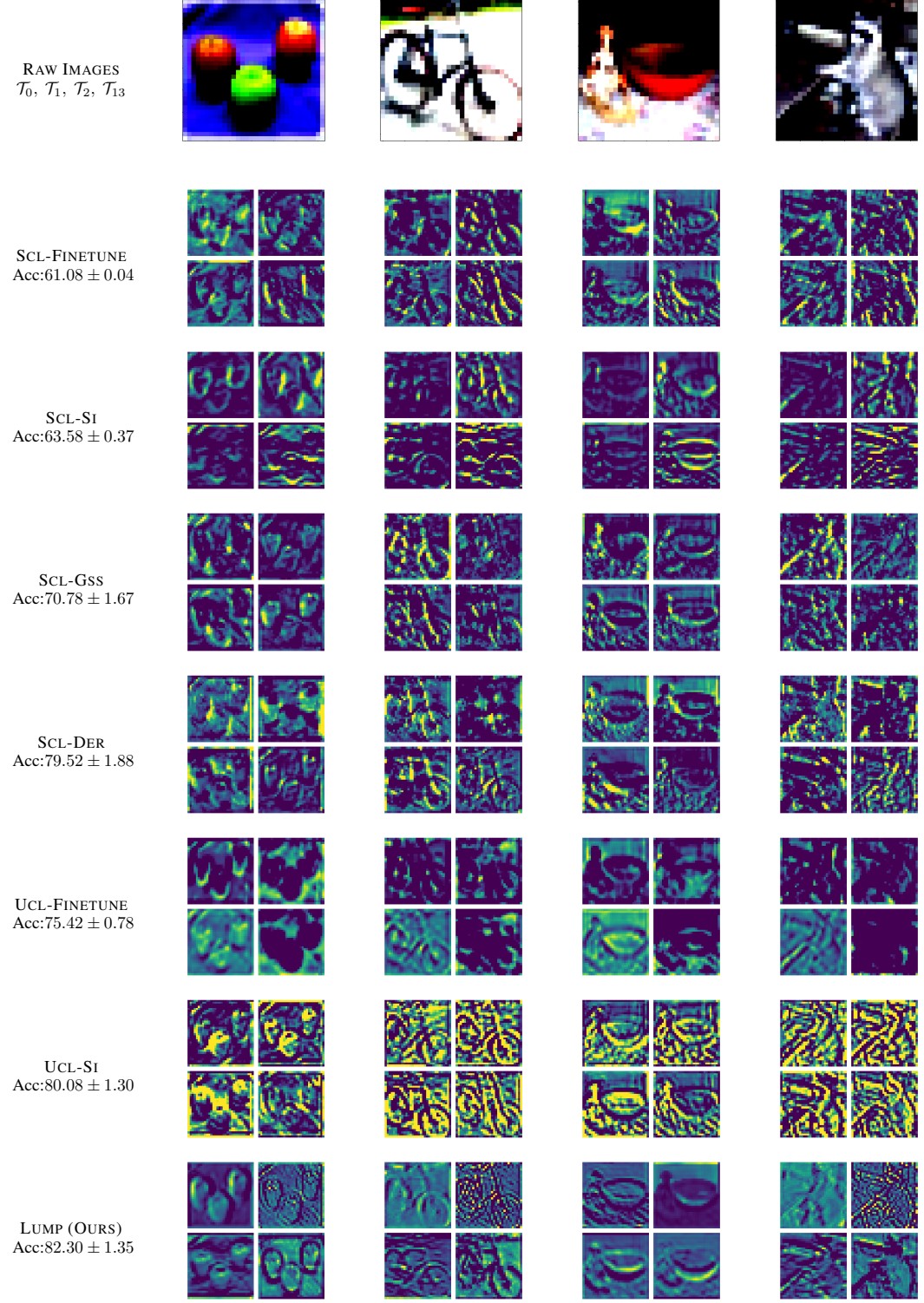

Figure A.7: **Visualization of feature maps** for the second block representations learnt by SCL and UCL strategies (with Simsiam) for Resnet-18 architecture after the completion of continual learning for Split CIFAR-100 dataset ($n = 20$). The accuracy is the mean across three runs for the corresponding task.

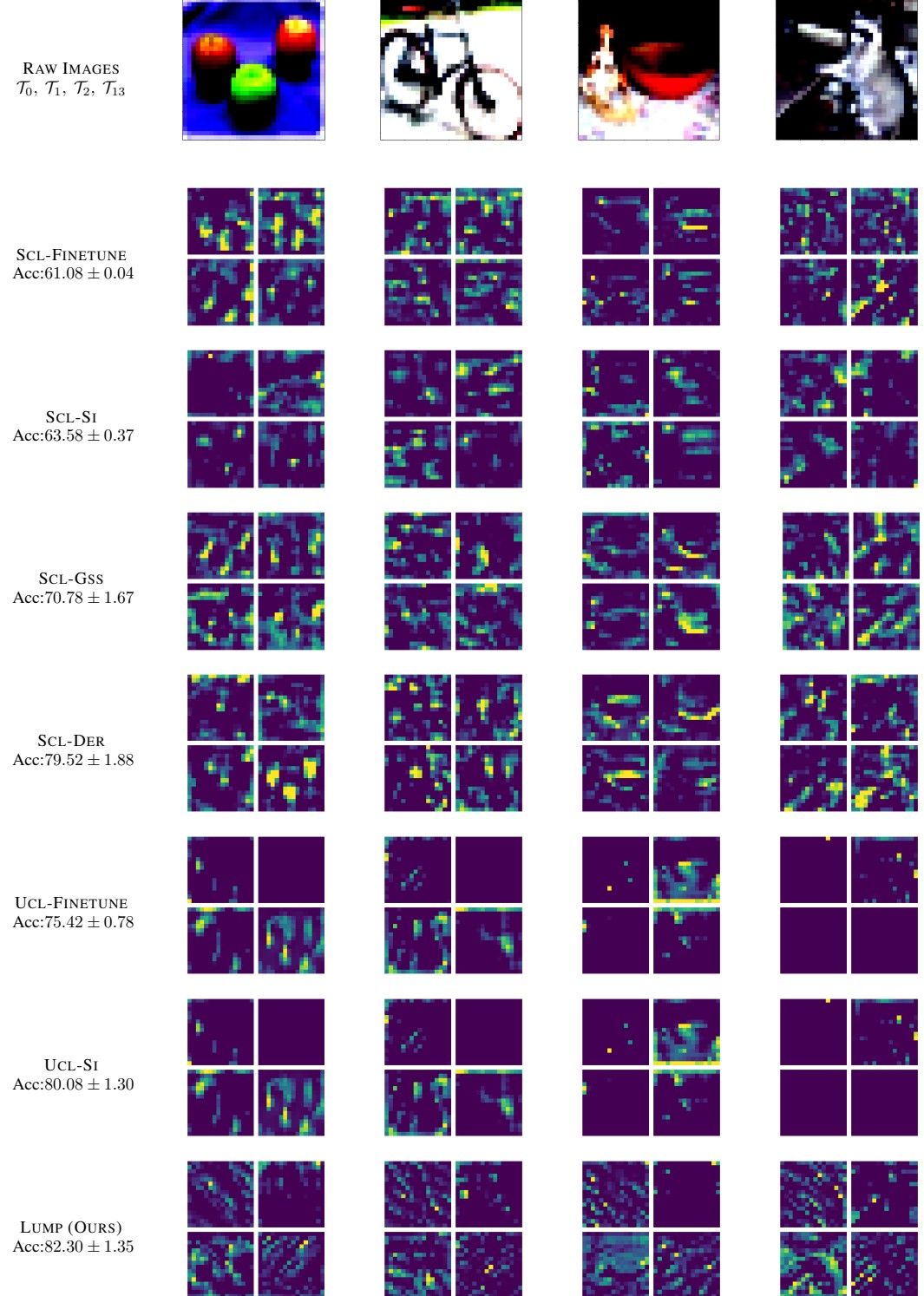

Figure A.8: **Visualization of feature maps** for the third block representations learnt by SCL and UCL strategies (with Simsiam) for Resnet-18 architecture after the completion of continual learning for Split CIFAR-100 dataset ($n = 20$). The accuracy is the mean across three runs for the corresponding task.

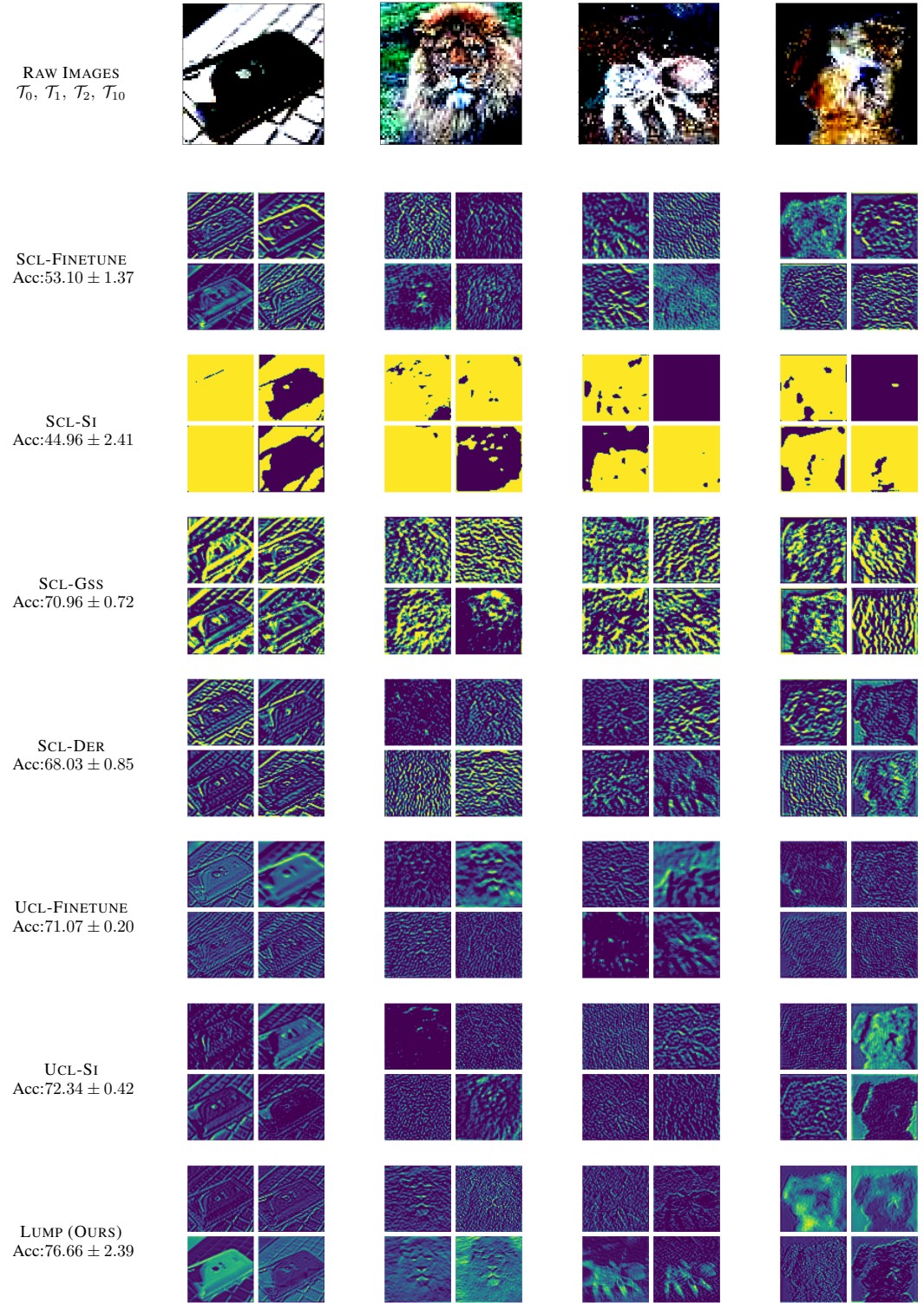

Figure A.9: **Visualization of feature maps** for the second block representations learnt by SCL and UCL strategies (with Simsiam) for Resnet-18 architecture after the completion of continual learning for Split Tiny-ImageNet dataset ($n = 20$). The accuracy is the mean across three runs for the corresponding task.

