# OpenReview forum: "Representational Continuity for Unsupervised Continual Learning"
_ICLR.cc/2022/Conference — ICLR 2022 Oral_

### Official Review · Reviewer_u1jR · 2021-10-29

**Correctness:** 3
**Technical Novelty And Significance:** 3
**Empirical Novelty And Significance:** 3
**Recommendation:** 8
**Confidence:** 3

**Main Review:**

### Strengths

* Rethinking continual learning with unsupervised representation learning is interesting, and empirical results indicate that most supervised continual learning methods can be improved by the proposed approach.

* A bunch of experiments have been conducted to demonstrate the effectiveness of the proposed approach in various settings. And several visualizations have also been included for a better understanding of the learned features.

### Weaknesses

* **[Missed Comparison with CURL]** \
Though the authors criticised that *Continual Unsupervised Representation Learning framework (CURL)* to be limited by digit-based gray-scale datasets, no direct comparison with CURL is done by following their evaluation protocol. Adding this result could better reveal the difference between the proposed method and CURL in terms of effectiveness. \
Beside, the evaluation of cluster quality used in CURL seems to be an important evaluation metric in unsupervised continual learning, which has not been used in the paper.

* **[Degraded Performance of DER and MULTITASK]** \
In the Table 1, we see that the proposed unsupervised continual learning can improve all baseline methods except DER and MULTITASK. A clear explanation about this performance drop should be added.

* **[Qualitative Analysis]** \
Why the visualization of feature maps stops at $\mathcal{T_{13}}$, while the loss landscape visualization continues to $\mathcal{T_{19}}$ ? And in Figure 5, the difference between SCL-DER and LUMP is hard to interpret.

* **[Why Not Directly Use Unsupervised Learned Presentations]** \
As an important purpose of the unsupervised representation learning is to learn a powerful embedding space that can be quickly fine-tuned for latter down-stream tasks. Why don't we consider a baseline where the feature backbone is initialized with SimSiam or Barlow Twins, and directly fine-tune them on a sequence of tasks. This is probably not considered in the standard continual learning, but the results of this baseline could be informative to the community of both domains.

**Summary Of The Paper:**

This paper attempts to bridge the gap between **unsupervised representation learning** and **continual learning** by extending various supervised continual learning methods to the unsupervised learning framework.
It builds upon two recent unsupervised feature learning techniques (*SimSiam* and *BarlowTwins*) and a powerful data augmentation technique *MixUp*.
Improved performances have been demonstrated in various experimental settings, together with comprehensive feature visualizations.

**Summary Of The Review:**

This paper attempts to rethink the standard continual learning in a new point of view by considering unsupervised representation learning methods. This purpose is interesting, but the major difference between the SCL (supervised continual learning) and UCL (unsupervised continual learning) seems to be just adding a unsupervised representation loss at the backbone and freezing it in the second stage of predicting head finetuning.
Moreover, some parts of the empirical results are not clearly presented or explained, an improved version of experimental results could be helpful for better validating the contributions.

---

> ### Author Response · Authors · 2021-11-17
> **Response to Reviewer u1jR - Part II**
>
> [see the previous comment for the first part of our response]
>
> > **Why Not Directly Use Unsupervised Learned Presentations.** As an important purpose of the unsupervised representation learning is to learn a powerful embedding space that can be quickly fine-tuned for latter down-stream tasks. Why don't we consider a baseline where the feature backbone is initialized with SimSiam or Barlow Twins, and directly fine-tune them on a sequence of tasks. This is probably not considered in the standard continual learning, but the results of this baseline could be informative to the community of both domains.
>
> Thank you for your suggestion. However, due to the limited compute budget and time during the rebuttal, we could not train an unsupervised model on ImageNet (powerful embedding space) to evaluate your proposed baseline. While the proposed baseline is orthogonal to both supervised and unsupervised continual learning scenarios, we believe it can be easily integrated with our model and further improve its performance.
>
> ---
>
> > The major difference between the SCL (supervised continual learning) and UCL (unsupervised continual learning) seems to be just adding a unsupervised representation loss at the backbone and freezing it in the second stage of predicting head fine-tuning.
>
> First, we want to clarify that our work is the *first attempt that bridges the gap between unsupervised representation learning and continual learning*. In particular, it allows the scalability of existing representation learning methods to a sequence of tasks and accommodates the *continuous shifts in data distributions* for representational learning. Further, our proposed framework is flexible and scalable that allows its application to multiple representation learning and continual learning methods, as demonstrated in our experimental evaluation.
>
> Second, note that UCL is simple, intuitive, easy to implement, and highlights the strengths of learning unsupervised representations. We show that a simple fine-tuning baseline can obtain comparable accuracy and lower forgetting than state-of-the-art supervised continual learning methods. We believe that our method will *open the door to understanding the behavior we demonstrate*, and we feel that it will be valuable for the community.
>
> Lastly, we provide comprehensive experiments to back up our claims and dissect the learned representations such as OOD evaluation, few-shot adaptation, features visualization, feature similarity, and loss-landscape visualization. We also demonstrate the utility of mixup for continual learning and propose a simple-yet-effective method coined LUMP that further *improves the performance and alleviates catastrophic forgetting*.
>
> ---
>
> Thank you again for your time and efforts in reviewing our paper, and we hope that you will consider raising your score if you find our response satisfactory.
>
> Thank you,
> Authors

---

> > ### Comment · Reviewer_u1jR · 2021-11-24
> > **Thx for reply**
> >
> > Thanks for your detailed response. All my issues have been well addressed, I thus increased the scores.

---

> ### Author Response · Authors · 2021-11-17
> **Response to Reviewer u1jR - Part I**
>
> Dear Reviewer u1jR,
>
> Thank you for your valuable feedback. We respond to the concerns raised by you below. Please let us know what you think about our response and whether you would like further clarifications.
>
> > **Missed comparison with CURL.**  Though the authors criticised that Continual Unsupervised Representation Learning framework (CURL) to be limited by digit-based gray-scale datasets, no direct comparison with CURL is done by following their evaluation protocol. Adding this result could better reveal the difference between the proposed method and CURL in terms of effectiveness.
> Beside, the evaluation of cluster quality used in CURL seems to be an important evaluation metric in unsupervised continual learning, which has not been used in the paper.
>
> We remark that the empirical evaluation CURL is limited to digit-based toy datasets (i.e., Split MNIST and Omniglot). The representations learned on these datasets are *not practically useful* for any downstream tasks in practical scenarios. Nevertheless, following the reviewer’s suggestion, we conducted experiments with SimSiam on Split-MNIST for task-incremental setting following CURL and presented the results below. We report the mean and standard deviation across three runs for accuracy and forgetting, while the CURL paper does not include the forgetting measure. Note that simply UCL-Finetune without any continual learning techniques can achieve *comparable performance* to CURL with LUMP, *further improving* the performance.
>
> |       		| Accuracy    	| Forgetting    	|
> |--------------		|------------------	|-------------------	|
> | CURL    		| $99.10$ $\pm$ $0.06$ 	| N/A       	|
> | UCL-FINETUNE 	| $98.16$ $\pm$ $0.32$ 	| $1.62$ $\pm$ $0.21$ 	|
> | UCL-LUMP  	| $99.27$ $\pm$ $0.17$ 	| $0.01$ $\pm$ $0.00$ 	|
>
>
> Additionally, we clarify that we report the accuracy and forgetting following the continual learning evaluation protocol. We utilize Wu et al. 2018 for our KNN evaluation and follow the KNN evaluation from prior representation learning methods, which is *more robust* than the CURL cluster accuracy/quality measure. In particular, the feature level embeddings learned by Wu et al. 2018 are *instance-wise discriminative*, while CURL measures the *class-discriminability* of latent-space. As described in Section A.4 of their paper, it is highly dependent on the dimensionality of the latent space. It works well for MNIST and Omniglot because they have many low-variance black pixels with a relatively small intra-class variance within the center pixels.
>
> ---
>
> > **Degraded Performance of DER and MULTITASK.** In the Table 1, we see that the proposed unsupervised continual learning can improve all baseline methods except DER and MULTITASK. A clear explanation about this performance drop should be added.
>
> We believe that the performance of DER degraded because DER was not formulated for UCL. In particular, our proposed approximation of utilizing the projected output instead of the logits in the original method might not the best possible choice and its investigation would be an interesting direction for future work.
>
> Further, MULTITASK shows a degraded performance because it shows the difference for supervised and unsupervised representation learning methods, where unsupervised methods do not outperform the supervised learning methods. In contrast, we show that unsupervised representations are surprisingly more robust to catastrophic forgetting when trained on a sequence of tasks.
>
> ---
>
> > **Qualitative Analysis.** Why the visualization of feature maps stops at $\mathcal{T}\_{13}$, while the loss landscape visualization continues to $\mathcal{T}\_{19}$?
>
> The reason that we provide the visualization of features at $\mathcal{T}\_{13}$ is to *demonstrate knowledge preservation of the task in the middle-time step* after the completion of continual learning. Of course, we can also provide the feature visualization of the last task, but it is not helpful to understand the catastrophic forgetting of feature representation during continual learning.
>
> In Figure 5, we visualize the loss landscape of *first task ($\mathcal{T}_0$)* after the completion of training task $\mathcal{T}_0$ and $\mathcal{T}\_{19}$, respectively, which helps to understand the change of loss landscape before and after occurring the catastrophic forgetting in a continual learner.
>
> ---
>
> > In Figure 5, the difference between SCL-DER and LUMP is hard to interpret.
>
> Figure 5, the difference between SCL-DER and LUMP is hard to interpret. We clarify that the scale of loss between SCL-DER and LUMP is different in Figure 5. We can observe that LUMP achieves a *lower value of loss with a smoother loss landscape* than SCL-DER. Further, we have provided other quantitative and qualitative analyses, including average accuracy and forgetting, feature visualization, feature similarity, and few-shot adaptation to demonstrate the effectiveness of LUMP compared to SCL-DER.
>
> ---
> [we continue our response below]

---

### Official Review · Reviewer_QCJy · 2021-10-29

**Correctness:** 4
**Technical Novelty And Significance:** 3
**Empirical Novelty And Significance:** 3
**Recommendation:** 8
**Confidence:** 4

**Main Review:**

Strengths:
- First the paper is well written and easy to read.
- The experiments are rich and well-chosen to better understand the superiority of unsupervised representations in the context of CL. I especially appreciated the experiments in Fig 2 that investigate the impact of the size of the training dataset.
- The conclusions are enlightening and will be very helpful to design new supervised or unsupervised CL methods.
- The code is publicly available and looks clean and  easy to use.

Weaknesses:
- Some details are unclear:
   - Sec 5.1: Knn classifier: which set is used for NN? the replay buffer? validation set?
   - The Knn is used both for supervised and unsupervised experiments, right?
   - Sec 5.3: more explanations about CKA are required for the reader that is not familiar with this measure
- I recommend changing the title. "Rethinking the Representational Continuity" is much too strong. The conclusions of the paper are great but it does not provoke a real rethinking of the problem.
- I am not really convinced but the visualization in Fig4. It seems that LUMP has sparser activations. The shapes of the objects are more clearly visible in its feature map. Does it simply mean that it learns lower-level features (similar to edge detector)? Maybe a TSNE visualization would help to see how the features of old tasks are affected when learning a new task.
- In Fig.5, we can notice that the range of value gets smaller in T19 (from [4.6,5.6] for T0 to [4.4,4.6]). Any idea why?




**Summary Of The Paper:**

The paper proposes to tackle the continual learning problem in an unsupervised setting. It shows that recent self-supervised learning methods are efficient tools to learn image representation with lower catastrophic learning problems. Two recent self-supervised methods are evaluated: SimSiam and BarlowTwins. In both cases, the superiority of unsupervised features is demonstrated.

The widely used mixup method is also adapted to the UCL problem. Straightforwardly, current images are mixed with images of the past tasks sampled from the replay buffer.

**Summary Of The Review:**

The paper shows interesting results that confirm the potential of self-supervised learning methods in the context of continual learning. The technical novelty may look incremental but the experimental conclusions are very interesting for the community. The paper is clear and well written.

---

> ### Author Response · Authors · 2021-11-17
> **Response to Reviewer QCJy**
>
> Dear Reviewer QCJy,
>
> We thank the reviewer for the encouraging comments and suggestions. We are glad that you found our paper well-written with rich experiments and believe that it will help design future supervised and unsupervised CL methods.
>
> ---
>
> > **K-NN classifier clarification.** Sec 5.1: Knn classifier: which set is used for NN? the replay buffer? validation set?
>
> We clarify that we use the K-NN evaluation proposed by Wu et al. 2018, where the features for each instance in the training set are stored in a discrete memory bank. The optimal feature-level embeddings are then learned by instance-level discrimination, which maximally scatters the features of the training samples. Following prior works in representation learning, we use the task-level training set without any augmentation in the task-incremental setup for the supervised and unsupervised KNN evaluation. We have further clarified this in Section A.1 of the revision.
>
> ---
>
> > The Knn is used both for supervised and unsupervised experiments, right?
>
> That is correct. We use KNN for evaluation for both the supervised and unsupervised experiments as described in Section 5.1.
>
> ---
>
> > Sec 5.3: more explanations about CKA are required for the reader that is not familiar with this measure
>
> Thank you for the suggestion. We have updated the description in the revision (Line 4 in Section 5.3, highlighted in blue).
>
> ---
>
> > In Fig.5, we can notice that the range of value gets smaller in T19 (from [4.6,5.6] for T0 to [4.4,4.6]). Any idea why?
>
> Thank you for pointing out this interesting observation. We believe that this is likely an outcome of the weight-decay during training. In particular, the scale of the network weights is *lower after the last task* compared to the first task. We hope our loss-landscape investigation would encourage future works to evaluate the loss-landscape for CL methods during sequential training to provide further insights to alleviate catastrophic forgetting.
>
> ---
>
> > I am not really convinced but the visualization in Fig4. It seems that LUMP has sparser activations. The shapes of the objects are more clearly visible in its feature map. Does it simply mean that it learns lower-level features (similar to edge detector)?
>
> We first want to emphasize that LUMP features in Figure 4 are not sparser (zero activations are visualized in dark blue). On the contrary, supervised continual learning counterparts and UCL-Finetune obtains more sparse and low-level features. Based on the following observations, we do not believe LUMP learns lower-level features like an edge detector: (a) it *significantly outperforms baselines* in both terms of accuracy and forgetting, which implies the *model preserves high-level feature information to distinguish the characteristics of each class* in the task which low-level features cannot perform, (2) visualized features from LUMP seem to capture the instance-discriminative knowledge while other baselines often obtain blurred and fragmented feature maps.
>
> We further include the visualization of the low-level features in Figure A.8 in the revision, where we obtain similar conclusions. In particular, UCL seems to capture *more localized low-level features* than SCL, which often captures redundant information. Note that the features in Figure A.8 are different from the features in Figure A.9, which capture high-level features with more information.
>
> ---
>
> > **Title update.** I recommend changing the title. "Rethinking the Representational Continuity" is much too strong. The conclusions of the paper are great but it does not provoke a real rethinking of the problem.
>
>  Thank you for raising your concern. We have updated the title to "Representational Continuity for Unsupervised Continual Learning" in the revision.
>
> ---
>
> Thank you,
> Authors

---

> > ### Comment · Reviewer_QCJy · 2021-11-29
> > **thanks for your answer.**
> >
> > Thanks for the detailed answer. My concerns are addressed. After reading the other reviews and the answers, I confirm my rating.

---

> > > ### Author Response · Authors · 2021-11-30
> > > **Thank you for your response**
> > >
> > > Dear Reviewer QCJy,
> > >
> > > We are glad to hear that we addressed your concerns. Thank you for all your insightful comments and suggestions.
> > >
> > > Thank you,
> > > Authors

---

### Official Review · Reviewer_tbme · 2021-10-31

**Correctness:** 4
**Technical Novelty And Significance:** 3
**Empirical Novelty And Significance:** 4
**Recommendation:** 8
**Confidence:** 4

**Main Review:**

Strong Points

* The paper takes one of the most import issues in CL: learning robust representation in unsupervised setting. For me, the problem itself is real and practical.

* The paper provides comprehensive experiments, including both qualitative analysis and quantitative results, to show the effectiveness of UCL and the proposed  LUMP algorithm over SCL methods.

* Overall, the paper is well written. In particular, the Related Work section has a nice flow and puts the proposed method into context. Despite the method having limited novelty, the method has been well motivated by pointing out the limitations in SOTA methods.

* The authors provide code for reproducing the results in the paper.


Weak Points

* The proposed LUMP algorithm is adopted from supervised mixup technique(Zhang et al, 2018). So the novelty is limited.

* The authors conducted extensive experiments in task-incremental setting. It would be interesting to see how UCL and the proposed method perform in class-incremental and task-agnostic CL settings.

* Fig 3: In general, higher layers have lower feature similarity than lower layers, and similarity between UCL models are higher than that of SCLs. However, there is an exception in Layer 4 of DER method -- the similarity of SCL is higher than that of UCL. It is worth some discussion on this exception.



**Summary Of The Paper:**

This paper studies the problem of representation learning in an unsupervised continual learning(UCL) setting. It shows that the representation learned with UCL is more general than the one learned with supervised CL (SCL), and investigates why UCL is more robust to catastrophic forgetting than SCL by analyzing the similarity of learned features and visualizing loss landscape. The authors also propose to apply mixup technique to UCL setting and present a LUMP algorithm to further improve the performance of CL.This paper studies the problem of representation learning in an unsupervised continual learning(UCL) setting. It shows that the representation learned with UCL is more general than the one learned with supervised CL (SCL), and investigates why UCL is more robust to catastrophic forgetting than SCL by analyzing the similarity of learned features and visualizing loss landscape. The authors also propose to apply mixup technique to UCL setting and present a LUMP algorithm to further improve the performance of CL.

**Summary Of The Review:**

Overall, I vote for marginally accepting. I like the idea of unsupervised continual learning and handling it by the proposed LUMP method. My major concern is about the limited novelty of the proposed method -- adopted from mixup in supervised learning, and some additional experiments on class-incremental or task-agnostic settings(see weakness above). Hopefully the authors can address my concern in the rebuttal period.

[After rebuttal]
The authors addressed most of my concerns, so I would like to raise my score.

---

> ### Author Response · Authors · 2021-11-17
> **Response to Reviewer tbme**
>
> Dear Reviewer tbme ,
>
> Thank you for your review and thoughtful comments. We are glad that you believe that our work is well-motivated, well-written, that tackles one of the most critical issues in CL with comprehensive qualitative and quantitative results. We clarified your concern regarding the novelty of our work and conducted additional experimental results showcasing the utility of UCL for class-incremental CL settings. We hope that you will consider raising your score if you find our response satisfactory.
>
> ---
>
> > **Novelty.** Limited novelty as LUMP algorithm is adopted from the supervised mixup technique(Zhang et al, 2018):
>
> We want to clarify that LUMP is not the main contribution of our work. In particular, our objective was to *analyze the UCL representations*, and in our experimental evaluation, we have shown that even without LUMP, UCL achieves *better performance* than its supervised counterparts. Further, we relax the assumption of the availability of a large amount of unbiased and unlabelled datasets to learn the unsupervised feature representations and propose *representation learning on a sequence of tasks*. To our knowledge, our work is the *first attempt that bridges the gap* between unsupervised representation learning and continual learning and scales UCL to larger datasets including TinyImageNet and Split CIFAR-100.
>
> To this end, we dissect this improvement by exhaustively comparing the SCL and UCL representations quantitatively on various datasets, OOD evaluation, few-shot adaptation, and qualitatively by feature visualization, feature similarity, and loss-landscape visualization. While mixup has not been utilized for continual learning in prior works, we use LUMP to improve the UCL counterparts, but we do not propose it as our main novelty.
>
> ---
>
> > **Class-incremental or task-agnostic CL experiments.** The authors conducted extensive experiments in task-incremental setting. It would be interesting to see how UCL and the proposed method perform in class-incremental and task-agnostic CL settings.
>
> Thank you for your suggestion. During the rebuttal, we compared various SCL and UCL representations for class-incremental CL scenarios. We present the results for accuracy in the class-incremental CL setup below. Note that UCL consistently outperforms the SCL counterparts. In particular, UCL-Finetune achieves an absolute improvement of *13.97% and 18.41%* over SCL-Finetune on Split CIFAR-10 and Split CIFAR-100, respectively. Further, UCL-SI improves the absolute performance over SCL-SI by *12.83% and 20.16%* on Split CIFAR-10 and Split CIFAR-100, respectively. We believe these results further demonstrate the flexibility of our method and strengthen our experimental results.
>
> |   	| Split CIFAR-10 	| Split CIFAR-100 	|
> |---------------	|------------------	|------------------	|
> | SCL-Finetune| $53.17 \pm 1.71$ 	| $13.77 \pm 0.12$ 	|
> | SCL-SI  	| $55.15 \pm 1.20$ 	| $15.77 \pm 0.61$ 	|
> | SCL-Multitask| $89.41 \pm 0.15$ 	| $66.07 \pm 0.18$ 	|
> | UCL-Finetune | $67.04 \pm 0.53$ 	| $32.18 \pm 0.53$ 	|
> | UCL-SI  	| $67.98 \pm 0.19$ 	| $35.93 \pm 2.33$ 	|
> | UCL-LUMP | $64.76 \pm 0.33$ 	| $38.84 \pm 0.42$ 	|
> | UCL-Multitask | $83.56 \pm 0.37$ 	| $51.05 \pm 1.60$ 	|
>
> ---
>
> > **Figure 3, an exception in Layer 4 of DER method.** Fig 3: In general, higher layers have lower feature similarity than lower layers, and similarity between UCL models are higher than that of SCLs. However, there is an exception in Layer 4 of DER method -- the similarity of SCL is higher than that of UCL. It is worth some discussion on this exception.
>
> We conjecture that this is due to the fundamental design of DER. In particular, SCL-DER matches the network logits across a sequence of tasks, which leads to the higher similarity for Layer 4. In contrast, this objective does not apply to UCL, and we match the output of the projected output, which might not be the best possible choice to retain all the properties of the supervised counterpart.
>
> Further, we want to clarify that it is essential to have higher similarity across lower layers because it highlights that the low-level representations learned across independently trained networks are robust and consistent. We believe that further investigation of this observation would provide more valuable insights for future work.
>
> ---
>
> We thank you again for your time and efforts in reviewing our paper and the constructive comments and suggestions.
>
> Thank you,
> Authors

---

### Official Review · Reviewer_synL · 2021-11-03

**Correctness:** 3
**Technical Novelty And Significance:** 3
**Empirical Novelty And Significance:** 3
**Recommendation:** 8
**Confidence:** 4

**Main Review:**

Strengths
* The submission is well written and easy to follow. The proposed concept is well motivated with various quantitative and qualitative justifications.
* While many unsupervised/self-supervised training approaches require pre-training on massive unlabeled data, the proposed method here works well with the help from Mixup and does not require additional pre-training set, largely making it more applicable to real-world use cases.
* I especially enjoy reading Sec 5.3 regarding the analyses on feature similarity between different learning approaches, visualization of feature space, and loss landscape visualization. This section provides additional justification besides the absolute accuracy improvement over the supervised continual learning counterparts.

Weaknesses
* I think the main limitation of using the proposed pipeline in practice is runtime and memory constraints. To run K-NN in a lifelong learning setting is challenging due to ever-growing storage requirements for storing samples coming from different stages. During inference, the algorithm also needs to compute distances between the query and many stored data points. Can the authors shed some light about the runtime and memory comparison between the proposed method and supervised counterparts?
* It would be great to also compare with other recent supervised continual learning approaches as well such as [A1, A2]

[A1] Zhao et al. Maintaining discrimination and fairness in class incremental learning. CVPR 2020. \
[A2] Liu et al. Mnemonics training: Multi-class incremental learning without forgetting. CVPR 2020.


**Summary Of The Paper:**

Unlike most of the continual learning approaches in the literature that perform supervised training at each learning stage, the authors propose to perform unsupervised representation learning on the sequence of incoming data and then classify the samples at each stage using K-NN. Experiments on standard CIFAR10/100 and Tiny-ImageNet shows that the proposed method alleviates catastrophic forgetting and generalizes better in different scenarios.

**Summary Of The Review:**

Overall a good quality submission with novel and interesting ideas. It would be great for the authors to address the weaknesses mentioned above.

---

> ### Author Response · Authors · 2021-11-17
> **Response to Reviewer synL**
>
> Dear Reviewer synL,
>
> Thank you for your review and your thoughtful comments. We appreciate you finding our paper well-motivated, well-written with quantitative and qualitative justification and applicability to real-world use cases. We clarify your concerns regarding computational constraints and comparison with class-incremental baselines below.
>
> > **Runtime and memory constraints for K-NN.** To run K-NN in a lifelong learning setting is challenging due to ever-growing storage requirements for storing samples coming from different stages. During inference, the algorithm also needs to compute distances between the query and many stored data points. Can the authors shed some light about the runtime and memory comparison between the proposed method and supervised counterparts?
>
> We utilize the K-NN method proposed by Wu et al. 2018, utilized in prior works for representation learning. It is computationally effective for large datasets and architectures. For instance, as mentioned in their paper, it can encode *1.28M ImageNet images in 600 MB* of storage, where exhaustive neighbor search takes 20 ms per image on a TitanX GPU. Similarly, we observed that it takes 110 seconds for the inference on the test set across all the tasks for Split CIFAR-10. The increase in computational cost is *marginal* compared to the supervised counterparts.
>
> ---
>
> > **Compare with other recent supervised continual learning approaches.** It would be great to also compare with other recent supervised continual learning approaches as well such as [A1, A2].
>
> Thank you for pointing us to these methods; however, we could not directly compare these methods during the rebuttal as the suggested methods focus on the class-incremental learning setup. In contrast, we conducted the experimental evaluation for the task-incremental setup. Nevertheless, we evaluated our method on class-incremental CL setup during the rebuttal, and we show the results below, where UCL achieves *significantly better performance* on both datasets. We report the mean and standard deviation across three independent runs.
>
> |       	| Split CIFAR-10 	| Split CIFAR-100 	|
> |---------------	|------------------	|------------------	|
> | SCL-Finetune| $53.17 \pm 1.71$ 	| $13.77 \pm 0.12$ 	|
> | SCL-SI    	| $55.15 \pm 1.20$ 	| $15.77 \pm 0.61$ 	|
> | SCL-Multitask| $89.41 \pm 0.15$ 	| $66.07 \pm 0.18$ 	|
> | UCL-Finetune | $67.04 \pm 0.53$ 	| $32.18 \pm 0.53$ 	|
> | UCL-SI    	| $67.98 \pm 0.19$ 	| $35.93 \pm 2.33$ 	|
> | UCL-LUMP  | $64.76 \pm 0.33$ 	| $38.84 \pm 0.42$ 	|
> | UCL-Multitask | $83.56 \pm 0.37$ 	| $51.05 \pm 1.60$ 	|
>
> ---
>
> Thank you,
> Authors

---

> > ### Comment · Reviewer_synL · 2021-11-26
> > **Thanks for the rebuttal**
> >
> > The authors have addressed my concerns. I will keep my score (8).

---

> > > ### Author Response · Authors · 2021-11-30
> > > **Thank you for your response**
> > >
> > > Dear Reviewer synL
> > >
> > > We are happy to hear that our response addressed your concerns. Thank you for your valuable feedback and suggestions.
> > >
> > > Thank you,
> > > Authors

---

### Decision · Program_Chairs · 2022-01-20

**Decision:**

Accept (Oral)

**Comment:**

Exciting work at the intersection of continual learning and representation learning. The reviewers have all commented that the proposed work addresses a number of issues related to catastrophic forgetting, which is very encouraging. The work also shows that the representation learning with the proposed method is more general than the one learned with supervised CL. The reviewers have praised the work as being well-written and with thorough experiments. There was a robust back and forth between the reviewers and the authors during the rebuttal period, in which the authors appear to have addressed most of the concerns. Given the insights, results and potential impact of this work, I think this work definitely should be published at ICLR.